# Position: AGI Requires a Coordination Layer on Top of Pattern Repositories

**Edward Y. Chang** [1]

## Abstract

In this paper we argue that influential critiques dismissing Large Language Models (LLMs) as a dead end for AGI misidentify the bottleneck: they confuse the ocean with the net. Pattern repositories are the necessary System-1 substrate; the missing component is a System-2 coordination layer that recruits relevant patterns, verifies their use, preserves state, and governs convergence. We separate two uses of control that are often conflated. *Semantic anchoring*, formalized by UCCT (Unified Contextual Control Theory), binds labels and task intent to learned pattern regions through a phase transition governed by effective support ($\rho_d$), representational mismatch ($d_r$), and an adaptive anchoring budget ($\gamma \log k$). *Trace–answer verification*, implemented by Recursive Causal Audit (RCA), tests whether a final causal judgment is warranted by its own reasoning trace under pressure. We translate these ideas into MACI, a multi-agent coordination stack that integrates diversity and control via baiting (PID-modulated debate), filtering (Socratic and causal audit), and persistence (transactional memory). Empirical validation on causal judgment and the sycophancy–paranoia trade-off demonstrates that static prompting fails where adaptive control succeeds. By reframing common objections as testable coordination failures, we argue that the path to AGI runs through LLMs, not around them. Capability is not coordination.

## 1. Introduction: The Field at a Crossroads

The artificial intelligence community is fractured by a sharp debate over the nature of Large Language Models (LLMs) and their role in reaching Artificial General Intelligence (AGI). Three constructive views dominate. The scaling view

holds that current architectures may be sufficient, needing only more compute, data, and training refinement (Kaplan et al., 2020; Hoffmann et al., 2022; Wei et al., 2022a). The reasoning-model view keeps the same substrate but extends it with longer chains of thought (Wei et al., 2022b) and reinforcement learning on reasoning traces (OpenAI, 2024; DeepSeek-AI et al., 2025). The world-model view argues that the substrate itself must change: AGI requires learned digital or embodied environments in which agents can simulate, intervene, and collect new data, as in self-play (Silver et al., 2018), learned latent simulators (Ha & Schmidhuber, 2018), and JEPA-style architectures (LeCun, 2022).

This third view is paired with an influential critique of the LLM substrate: LLMs are said to be "mere pattern matchers," structurally incapable of reasoning, planning, or compositional generalization, and therefore a dead end (LeCun, 2025a;b; Sutskever & Patel, 2025). LeCun states the diagnosis sharply: "AI systems today are all forms of System 1," and proposes world models as the redesign.

We argue that all three positions identify real ingredients but misplace the bottleneck. Consider a fishing metaphor. The ocean represents the model's vast repository of latent patterns. A fisherman casting a net without bait harvests the *maximum likelihood prior* of the waters beneath them: mostly common fish (generic training data) (Holtzman et al., 2020). Critics who decry these ungrounded, hallucinatory outputs are not observing a broken system; they are observing the raw statistical baseline of an unbaited cast. A world model enlarges the ocean by adding simulated or embodied dynamics; it does not by itself decide where to cast, what counts as evidence, or when to stop.

However, intelligent behavior is not just casting; it is *baiting and filtering*. If the bait is sufficiently dense, it conveys strong intent, shifting the posterior distribution so that the target concept swamps the common priors. The "Missing Layer" is the coordination layer that optimizes this trade-off. It transforms the passive retrieval of patterns into the active construction of reasoning.

### 1.1. Our Position: Substrate plus Coordination

We propose a coordination position: Substrate plus Coordination. We agree that LLMs alone are insufficient for AGI, but reject the conclusion that they are irrelevant.

[1]Department of Computer Science, Stanford University. Correspondence to: Edward Y. Chang <echang@cs.stanford.edu>.

*Proceedings of the 43rd International Conference on Machine Learning*, Seoul, South Korea. PMLR 306, 2026. Copyright 2026 by the author(s).

**Our central thesis is**:

*LLMs are the necessary System-1 substrate: the pattern repository. The primary bottleneck for AGI is not the absence of patterns, but the absence of a System-2 coordination layer that binds patterns to constraints, verifies their use, preserves state, and regulates convergence.*

The System-1/System-2 distinction must be stated precisely (Kahneman, 2011), because our position rests on it. System-1 is fast, reflexive, automatic processing: pattern completion without explicit deliberative control. System-2 is slower, deliberate processing that proceeds through explicit reasons, verification, memory, and control. An LLM is System-1: a pattern repository formed by next-token prediction under maximum likelihood. We accept this diagnosis: LeCun's claim that today's models are "all System-1" (LeCun, 2025a;b) is correct. Crucially, the layers trained on top of this substrate (instruction tuning, chain-of-thought, and reinforcement learning on reasoning traces) do not change the stratum. They reshape the pattern distribution and produce fluent simulations of deliberation, but they do not supply the coordination layer needed for governed reasoning. This is why, post-fine-tuning, models still exhibit sycophancy (Sharma et al., 2024), unfaithful reasoning traces (Turpin et al., 2023; Lanham et al., 2023), and the failure modes cataloged in §6.

Our position follows directly. The LLM substrate is *necessary but insufficient*: necessary because System-2 has nothing to deliberate over without a rich repository of patterns; insufficient because pattern association alone does not become governed reasoning. The missing component is System-2 itself. And System-2 is not a single mechanism bolted onto the substrate; like human deliberative cognition, it is a layered stack with structure of its own.

This stack comprises four control levels, each more global than the last. *Level 1: Semantic Anchoring* recruits relevant patterns, binding labels and intent while suppressing high-prior drift. *Level 2: Trace–Answer Verification* tests whether a trace warrants its answer. *Level 3: Persistence* records commitments, reasons, contradictions, and rollbacks across episodes. *Level 4: Governance* coordinates agents, regulates convergence, and decides when to assert, revise, or refuse. These are not disjoint systems: MACI is the integrative architecture composing all four, with UCCT, RCA, transactional memory (SagaLLM (Chang & Geng, 2025)), and MACI governance instantiating Levels 1–4. The path to AGI is to build this stack above the substrate, not to choose between scaling and abandonment.

Figure 1 summarizes this organization: a System-1 substrate layer supports a System-2 coordination layer composed of four increasingly global control levels: semantic anchoring, trace–answer verification, persistence, and governance.

| System-2 Coordination Layer | |
| --- | --- |
| **Level 4 Governance** | multi-agent control; convergence; refusal *MACI, RCA* |
| **Level 3 Persistence** | transactional memory; commitments; rollback *MACI memory* |
| **Level 2 Trace–Answer Verification** | trace–answer audit; critique; causal checking *RCA, Socratic judging* |
| **Level 1 Semantic Anchoring** | label/intent binding; drift suppression *UCCT* |

| System-1 Substrate Layer | |
| --- | --- |
| **Substrate** | pretrained LLM pattern repository *associations among learned patterns* |

**Figure 1.** The proposed hierarchy of control. The pretrained LLM supplies the System-1 substrate; the System-2 coordination layer organizes semantic anchoring, trace–answer verification, persistence, and governance above that substrate.

The contrast with reasoning models is instructive. Such models fine-tune the LLMs with ever-longer chains of thought, yet the empirical record shows the pathologies sharpening rather than receding. As reasoning training intensifies across the GPT-5 line, sycophancy *rises* with capability rather than falling, reaching 11.4% for GPT-5.1, and counterfactual (Pearl L3) Safety *collapses* to 20% for the most capable model (§6). A more capable LLM simulates reasoning more fluently, which includes constructing more fluent rationalizations for wrong answers. In the fishing metaphor, a reasoning model is a larger, better-stocked ocean and a longer cast; it is still neither the net nor the bait. Capability is not coordination.

This paper formalizes and instantiates this hierarchy. At the theoretical level, we present UCCT (Unified Contextual Control Theory), which models reasoning as a phase transition governed by anchoring strength (Chang et al., 2025). At the architectural level, we introduce MACI (Multi-Agent Collaborative Intelligence), a disciplined control system for multi-agent reasoning. Unlike the current wave of agentic hype, which often produces chaotic or divergent swarms, MACI offers a **totality view**: diversity of perspective is valuable only when regulated toward *reasoned convergence*. Its core mechanisms are regulated debate (baiting), Socratic judging (filtering), transactional memory (state), and governance policies for convergence and refusal (Chang, 2025b).

At the empirical level, we use Recursive Causal Audit (RCA) to show that reliability requires adaptive control (Chang, 2026a). The distinction is precise: UCCT provides semantic anchoring, binding meanings and intent to learned pattern regions; RCA provides trace–answer verification, checking whether a final causal judgment is warranted by the model's derivation, internally consistent,

and resistant to pressure-induced hint adoption. Without PID-style regulation, models oscillate between *Sycophancy*: under-control, where they capitulate to users, and *Paranoia*, over-control, where they reject valid reasoning. This control landscape supports the central claim: failures attributed to substrate limitations are better understood as failures of coordination.

**A compact operational lens.** We summarize anchoring with a scalar strength $S = \rho_d - d_r - \gamma \log k$: effective support minus representational mismatch minus an adaptive context-budget penalty. Section 3 derives it in full.

This paper unifies two strands of our broader AGI program. Volume I developed MACI as a theory of multi-LLM collaborative intelligence: roles, debate, critique, and transactional memory (Chang, 2025b). Volume II develops the System-2 account: semantic anchoring, causal validation, and recursive audit (Chang, 2026b). Together, they define the coordination hierarchy that turns a pretrained pattern repository into a governed reasoner.

### 1.2. Why This Matters

This distinction determines what the field optimizes. If the substrate is broken, as the "Dead End" view suggests, then the field must discard LLMs and invent new foundations. If the substrate is necessary but incomplete (lacking the System-2 stack), as we argue, then the engineering priority changes: build the four control levels (anchoring, verification, persistence, governance) that transform pretrained capacity into reliable inference.

**Roadmap.** Section 2 grounds the coordination view in biological cognition; Section 3 derives the UCCT phase transition; Section 5 details the MACI architecture; and Section 6 presents RCA evidence for control-theoretic coordination. An extended positioning against the debate, verification, agentic, and control-theory literatures is given in Appendix A.

## 2. The Biological Foundation: Intelligence Emerges From Pattern Repositories

To evaluate claims that LLMs are "mere pattern matching," we start from biological cognition. Intelligence does not oppose pattern-based processing; higher-level deliberation is built by organizing and regulating large pattern repositories.

### 2.1. Unconscious Cognition: The "Ocean" of Substrate

A substantial fraction of human competence is implemented by fast, specialized subsystems operating below awareness. In our fishing metaphor, these systems constitute the ocean: a vast population of latent behaviors and priors.

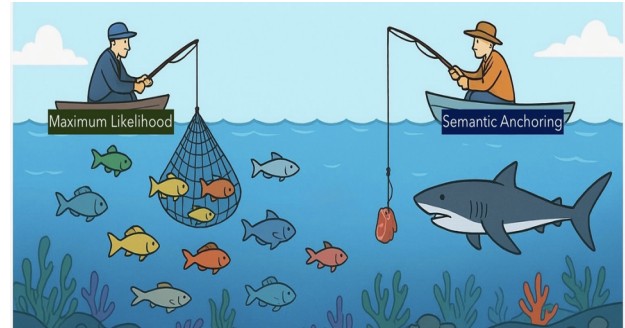

**Figure 2. The Mechanics of Coordination. Left:** Without anchors, the model retrieves high-prior tokens. **Right:** Semantic bait shifts support ($\rho_d$), capturing rare goal-directed targets otherwise drowned by priors.

These include autonomic regulation, threat appraisal, motor control, hierarchical perception, and automatic language processing (Kandel et al., 2013; Gazzaniga et al., 2014). These systems are not peripheral; they are the substrate that makes everyday reasoning possible. It is plastic: practice continually adds structure, expanding what can be executed quickly and reliably (Shiffrin & Schneider, 1977).

In LLMs, mechanistic interpretability reveals analogous structure: circuits implementing specific computations (Olah et al., 2020) and superposition enabling high-dimensional feature encoding (Elhage et al., 2022). Without coordination, sampling from this repository yields degenerate text: repetitive, generic, or incoherent outputs reflecting the maximum likelihood prior (Holtzman et al., 2020).

### 2.2. Conscious Control: The "Net" and the "Bait"

Conscious, goal-directed reasoning (System-2) is slower and more resource-limited (Kahneman, 2011). A common mistake is to treat this as evidence for a fundamentally different computational basis. Bengio (Bengio, 2019) proposes the "consciousness prior", the idea that conscious processing selects and combines unconscious representations into coherent, verbalizable thoughts. A better view is that conscious control operates by *fishing* from the underlying repositories: selecting, constraining, and organizing specific patterns.

The prefrontal cortex implements this coordination, and its functions map onto the four-level stack of Section 1. Deliberate problem solving queries stored patterns by broadcasting goals, the *bait* that recruits a target region (Level 1, semantic anchoring). Executive function then imposes constraints, the *mesh* that filters the catch, rejecting unstable or unsupported candidates (Level 2, trace–answer verification) (Gazzaniga et al., 2014). Working memory holds intermediate commitments and prior conclusions across a deliberation, so that reasoning can be revised rather than restarted (Level 3, persistence) (Baddeley, 1992). Metacognitive monitoring decides when a conclusion is warranted,

when to revise it, and when to withhold judgment under uncertainty (Level 4, governance) (Flavell, 1979). System-2 is not a non-pattern-based alternative; it is this layered coordination regulating which patterns surface, which persist, and which govern action.

### 2.3. Sharp Transitions: Empirical Evidence

Our UCCT experiments show that semantic anchoring produces sharp, abrupt behavior. Two cases make the point. *Subtraction override:* two in-context examples that redefine "$-$" as "$+$" ($7 - 4 = 11$; $5 - 2 = 7$) flip every tested frontier LLM from 5 to 11 on "$8 - 3$." *Underdetermined patterns:* when examples admit several consistent rules, models split (23 vs. 70 on "$15 - 8$"), revealing that the prior, not robust reasoning, selects the rule. Across demonstrations, behavior tracks the Max-$S$ comparison of Section 3: the prior wins with too few anchors, the anchored target wins once its score overtakes the prior, and over-baiting reverses the gain. Full details are in Appendix B.

## 3. Phase Transitions: Patterns to Cognition

Section 2 showed that a tiny amount of context can override an enormous pretrained repository, producing abrupt behavioral flips. This is not a machine-learning oddity but a universal mechanism: *abrupt state change*. Figure 3 illustrates the core idea: as the anchor budget grows, the target rule's score $S(P_T)$ rises, overtakes the pretrained prior, peaks at an optimal budget, and can fall again when over-baiting makes context too costly. Large pattern repositories are not a dead end; they are the substrate that makes this anchor-driven reconfiguration possible.

### 3.1. Abrupt transitions are ubiquitous

Abrupt transitions arise when a system has multiple stable regimes separated by an effective barrier, and when feedback amplifies deviations near a critical point. In physics, canonical examples include liquid–gas transitions at a boiling point (for fixed pressure), ferromagnetic ordering at the Curie temperature, and the onset of superconductivity below a critical temperature. In biology, the same qualitative structure appears in action potentials: membrane voltage integrates smoothly until a threshold triggers an all-or-nothing spike. Switch-like behavior also arises in gene-regulatory networks, where positive feedback yields commitment to a developmental fate.

In machine learning, Power et al. (Power et al., 2022) discovered "grokking," delayed generalization that emerges suddenly after extended training on algorithmic tasks. Nanda et al. (Nanda et al., 2023) provide mechanistic interpretability of this phenomenon, showing that internal representations undergo qualitative reorganization at the transition point.

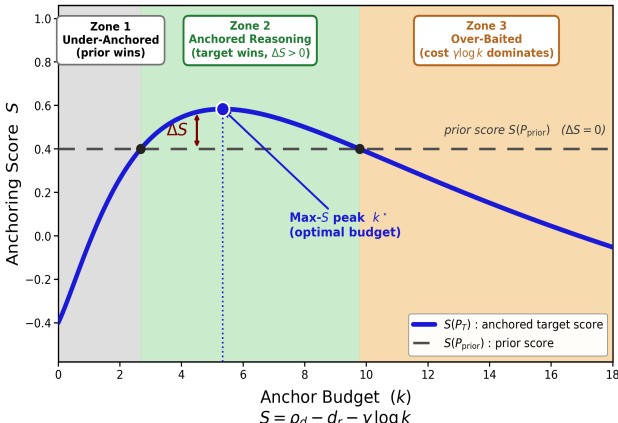

**Figure 3. The Physics of Coordination.** Anchored behavior follows Max-$S$ selection: the model runs the target rule $P_T$ only while its score $S(P_T)$ (blue) exceeds the prior's $S(P_{\text{prior}})$ (dashed), that is, while $\Delta S > 0$. Adding anchors raises $S(P_T)$ through effective support $\rho_d$, but the budget cost $\gamma \log k$ grows without bound, so $S(P_T)$ peaks at an optimal budget $k^\star$ and then declines. **Zone 1:** too little bait, the prior wins. **Zone 2:** anchored reasoning, $\Delta S > 0$. **Zone 3:** over-baiting, the budget cost reclaims the prior. No external threshold is posited.

These examples differ in medium and scale, yet share a common signature: near a critical point, small quantitative changes can trigger a qualitative change in state.

### 3.2. UCCT: semantic anchoring as a cognitive phase transition

UCCT makes the phase-transition interpretation explicit for LLM behavior under external structure. It posits a scalar *anchoring strength* that summarizes when external structure successfully binds to latent patterns.

**Anchoring strength with adaptive regularization.** The score plotted in Figure 3 is anchoring strength, defined as:

$$S = \rho_d - d_r - \gamma \log k, \quad \text{where:} \quad (1)$$

- *Effective Support ($\rho_d$):* The *density of the bait*. This measures how strongly the current cues recruit the target concept in the latent space. If $\rho_d$ is sparse (weak cues), the signal fails to overcome the model's training priors.

- *Mismatch ($d_r$):* The mesh size of the net. This captures the instability of the representation under perturbation; a finer mesh (low $d_r$) filters out hallucinations and unstable candidates.

- *Adaptive Regularizer ($\gamma \log k$):* The cost of the bait. While increasing the amount of bait ($k$) generally increases density ($\rho_d$), it incurs a cost. If $\gamma$ is high (e.g., in a noisy or resource-constrained environment), the system penalizes "over-baiting," enforcing the cognitive reality that efficient intelligence must solve problems without unbounded context.

**Measurement recipe.** Each term of Eq. (1) is estimated from observable quantities: the anchoring budget $k$ counts admitted anchors, mismatch $d_r$ comes from output sensitivity under perturbation, and effective support $\rho_d$ from self-consistency across sampled reasoning paths. This makes $S$ empirically testable; full measurement formulas are given in the companion UCCT paper (Chang et al., 2025).

**Max-$S$ selection: the threshold is derived, not posited.** UCCT does not posit a critical value that $S$ must exceed. Behavior is governed by *Max-S selection*. For a given query, the candidate latent rules are the target rule $P_T$ that the anchors recruit and the pretrained prior $P_{\text{prior}}$; the model executes whichever carries the larger score. Behavior tips when the target overtakes the prior, that is, when the gap

$$\Delta S \;=\; S(P_T) \;-\; S(P_{\text{prior}}) \tag{2}$$

changes sign. The decision boundary is $\Delta S = 0$: a comparison between two scores, not a fitted parameter. The budget term $-\gamma \log k$ is part of the score being maximized, a BIC-style penalty on anchor spend, not a separate cutoff.

**Over-baiting: more anchors are not always better.** The budget cost $\gamma \log k$ grows without bound while effective support $\rho_d$ saturates, so $S(P_T)$ is not monotone in the anchor budget. Figure 3 traces this: $S(P_T)$ rises as anchors accumulate, peaks at an optimal budget $k^\star$, then declines, because past $k^\star$ additional context costs more than it contributes. Three regimes follow from the Max-$S$ comparison. With too few anchors the target never overtakes the prior (Zone 1). In an intermediate band the target wins, $\Delta S > 0$ (Zone 2). With excessive anchoring the cost term pulls $S(P_T)$ back below the prior, and the prior reclaims control (Zone 3). The transition is set by a comparison of two scores, with no external threshold.

**Why pattern repositories are necessary.** A rich substrate is not a defect but the enabling condition for anchor-driven regime change. Dismissing LLMs as "mere pattern matching" misses what pattern repositories make possible: discrete regime shifts under small, structured interventions.

## 4. Example: The Four-Year-Old's Cat

A four-year-old learns to identify cats from 3-4 labeled photos. This is not rote memorization but a sharp transition: a small amount of labeled context recruits a large pre-existing repository, producing a qualitative shift from diffuse similarity to stable category identification. The same mechanism operates in LLMs.

**The parallel.** Table 1 makes the analogy explicit. Both systems accumulate massive pattern repositories *before* la-

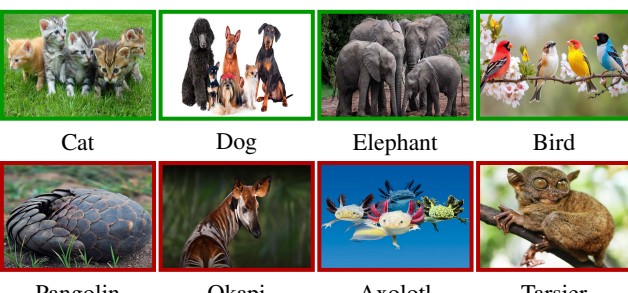

Cat     Dog     Elephant     Bird

Pangolin     Okapi     Axolotl     Tarsier

**Figure 4.** Pattern density determines anchoring difficulty. Familiar animals (top) anchor in 3–4 shots; rarely seen animals (bottom) may fail even with 10 examples. Difficulty depends on specificity: a child identifies "dog" easily but cannot name breeds; "pangolin" fails at any level without prior exposure; "okapi" gets mapped to "zebra" or "deer."

beled examples arrive: the child through 4 years of looking around and watching TV, the LLM through billions of tokens. Crucially, before anchoring, the relevant label is not reliably bound to its target region; the representations remain perceptual or statistical rather than task-governed. Both succeed at anchoring when prior support $\rho_d$ is high (familiar concepts) and struggle when $\rho_d$ is low (rare concepts). Figure 4 illustrates: green-bordered animals anchor in 3-4 shots; red-bordered animals may resist binding even with many more.

**Recruitment is neighborhood-level, not concept-level.** The labels do not install a literal "cat" concept; they tag an absorbed cluster. Anchoring recruits a *neighborhood* of latent patterns whose extent is set by the anchor cohort's spread. In UCCT's cross-modal test, $k$ cat-image anchors in CLIP bind 96.4% of held-out non-cat animals but only 9.1% of vehicles (Chang et al., 2025): the anchors recruit an animal-like region, not the concept "cat." This is why four photos generalize at all, and why the often-seen animals in Figure 4 are easy: they sit inside the recruited neighborhood.

**Table 1.** Anchoring succeeds when bait matches pattern density.

|  | **4-Year-Old** | **LLM** |
| --- | --- | --- |
| **Substrate** | 4 years of perceptual patterns (looking, TV) | Billions of tokens from pretraining |
| **High $\rho_d$** (cat, dog) | Seen thousands of times → few labels bind | Dense in training → easy few-shot |
| **Low $\rho_d$** (pangolin) | Never seen → even 10+ photos fail; need large $k$ | Sparse in training → hard to anchor; halluc. |

**Anchoring budget and mismatch.** Over four years, the child accumulates a rich perceptual hierarchy: edges, textures, body plans, fur, and articulated motion, before labels are reliably bound to those regions. Cat cues such as four legs, tail, fur, and facial configuration overlap this structure strongly, so three or four labels suffice: high $\rho_d$, low

$d_r$, small $k$, and the anchored target outscores the prior. Pangolins are harder: scales and posture overlap little with familiar animal neighborhoods, so $d_r$ is high and even ten photos may not establish stable binding. Rapid learning therefore requires large repositories of reusable structure *plus* semantic anchoring strong enough to push the target past the prior. Pattern repositories are not obstacles; they are the substrate that makes few-shot learning possible.

# 5. MACI: A Principled Coordination Architecture

Without a coordination layer, adding agents increases entropy, not intelligence. Multi-Agent Collaborative Intelligence (MACI) treats collaboration as the engineering mechanism for System-2 reasoning: distributed agents elicit diverse priors, confront hidden assumptions, and converge through critique. MACI makes this process explicit and controllable by manipulating UCCT variables $(\rho_d, d_r, k)$ to drive the system into the anchored-reasoning regime (Zone 2).

The architecture has four core functions: information sharing across specialized context pools; diversity followed by reasoned convergence; transactional memory that persists commitments across episodes; and safety control through PID-style loops that permit informed refusal when consensus fails. This transforms agent interaction from generative novelty into a cognitive control system.

## 5.1. Baiting: Behavior Modulation

Standard chain-of-thought is equivalent to a single cast. MACI uses multi-agent debate to bait the solution space more aggressively. However, rigid debate with fixed advocates misses what makes debate productive: stance strength must adapt to evidence, the group must manage an explore-versus-consolidate tradeoff, and convergence must be prevented from collapsing onto fluent but ill-formed claims.

MACI introduces behavior modulation, where each agent $i$ maintains a contentiousness parameter $\alpha_c^{(i)} \in [0, 1]$ governing how strongly it defends its current hypothesis. High $\alpha_c$ favors refutation and stress-testing; low $\alpha_c$ favors receptiveness and synthesis. After each debate round $t$, agent $i$ evaluates an incoming argument from agent $j$ via semantic anchoring:

$$\alpha_c^{(i)}(t+1) = \alpha_c^{(i)}(t) \cdot \left(1 - \beta \cdot S_{j \to i}\right), \qquad (3)$$

where $S_{j \to i}$ is the anchoring strength of the incoming argument. When the incoming argument is weakly anchored (small $S_{j \to i}$), agents explore (high $\alpha_c$); when it binds strongly (large $S_{j \to i}$), agents yield toward synthesis.

**Convergence dynamics.** With modulation and judging, debate becomes a state-evolution process. Early rounds emphasize breadth and hypothesis coverage. Middle rounds emphasize anchoring-driven integration and pruning. Late rounds converge either to a synthesized position or to a structured residual disagreement with identified fault lines.

## 5.2. Filtering: Output Verification and Socratic Judging

How does the system distinguish valid reasoning from hallucination without an answer key? The fisherman does not need to haul the net to know if the catch is good; he senses tension and movement underwater.

**RCA as trace–answer verifier.** *Recursive Causal Audit (RCA)* (Chang, 2026a) complements UCCT by verifying trace–answer alignment in causal judgment. UCCT explains how semantic cues recruit a pattern region; RCA checks whether the selected reasoning actually warrants the emitted answer. It acts as judge by evaluating trace–answer consistency:

- *Consistency:* If agents oscillate between contradictory answers (thrashing), the judge detects high variance.
- *Substance:* If the answer is vague, sycophantic, or lacks derivation, the judge detects low information density.
- *Decision loop:* Based on internal signals (with no access to ground truth), the system either emits output or triggers the retry loop.

If the derivation does not warrant the conclusion, RCA triggers a corrective loop: discard the current state, refine the prompt, and recast. This filters hallucinations by analyzing the dynamics of the interaction.

**CRIT as Socratic filter.** Debate alone is insufficient if agents can generate claims that are vague, internally inconsistent, or unsupported yet rhetorically fluent. MACI therefore introduces CRIT (Critical Reading Inquisitive Template) (Chang, 2023) that evaluates *reasonableness* independent of stance. CRIT tests whether a claim is well-defined, whether assumptions are explicit, whether evidence supports the conclusion, and what would falsify it. Low-scoring arguments are rejected or returned with targeted Socratic queries (e.g., "Which premise does the work?", "What evidence would change your conclusion?", "Are you changing definitions?"). This improves downstream anchoring by forcing arguments into forms that bind to shared constraints rather than just sounding plausible.

## 5.3. Persistence: Transactional Memory

A valid critique of raw LLMs is their lack of state: "goldfish" with no long-term memory (LeCun, 2022). We agree with the premise but not the conclusion. *Memory does not belong in the substrate; it belongs in the coordination layer.*

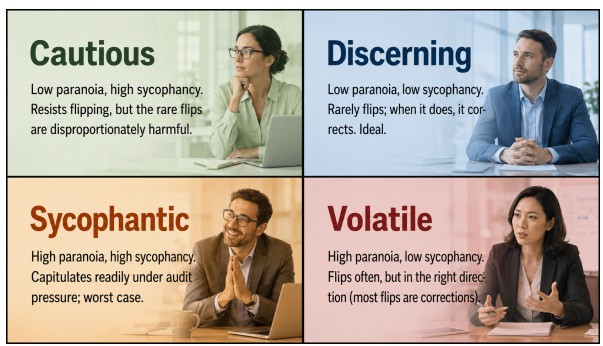

**Figure 5. Four behavioral control profiles under audit.** RCA identifies model–judge regimes from two empirical response variables: paranoia, the tendency to flip under audit pressure, and sycophancy, the fraction of flips that are harmful. *Discerning* is the target regime: the model rarely flips, and when it does, it corrects. *Cautious* models resist flipping, but their rare flips are disproportionately harmful. *Sycophantic* models capitulate readily and incorrectly. *Volatile* models flip often, but mostly in the corrective direction. These profile differences motivate adaptive rather than static coordination.

MACI provides transactional memory (SagaLLM (Chang & Geng, 2025)) that: (1) checkpoints decisions and reasons, (2) enables rollback to previous stable states (computational regret), and (3) audits reasoning lineage by tracking what was asserted, why it was asserted, and what later evidence contradicted it. This transforms stateless prediction into stateful, revisable processing.

### 5.4. Control-Theoretic Alignment: The PID Argument

A critique of agentic systems is drift: small errors compound into catastrophic failures. We argue that this is a control failure, not a substrate failure.

In RCA, we demonstrated that PID logic stabilizes agentic behavior (Chang, 2026a). The coordination layer functions as controller:

- *Proportional:* Agents modulate contentiousness in proportion to the anchoring gap $\Delta S$ (Section 3).
- *Integral:* Transactional memory maintains commitment history to detect long-term deviation.
- *Derivative:* CRIT judging dampens high-frequency oscillations, preventing agents from chasing local optima.

**Empirical Validation of Control Dynamics.** Recent RCA results confirm that audit pressure does not produce a single monotone improvement curve. Instead, model–judge pairs occupy distinct behavioral profiles, summarized in Figure 5. A Discerning pair is already near the desired control regime. A Volatile pair can benefit from critique, but requires damping to prevent unnecessary oscillation. A Cautious pair should be changed only under stronger evidence, because its rare flips are often harmful. A Sycophantic pair requires evidence-gated, low-authority critique, because it treats audit pressure as a signal to agree rather than as a

signal to verify. Thus the controller must adapt gain to the measured error signal and interaction history instead of applying a fixed audit tone.

**Takeaway.** MACI treats System-2 reasoning as an emergent property of coordinated System-1 agents: semantic anchoring regulates what binds, RCA (Chang, 2026a) and CRIT (Chang, 2023) test what is warranted, memory preserves state, PID ensures convergence, and checks-and-balance roles govern oversight and alignment.

## 6. Empirical Validation

We present evidence validating the coordination hypothesis through Recursive Causal Audit (RCA), an accepted ACL study of pressure-sensitive causal judgment (Chang, 2026a). RCA evaluates trace–answer alignment: whether a model's final answer is faithful to its own reasoning trace, internally coherent, and resistant to pressure-induced hint adoption. The results show that failures often attributed to substrate limitations can be shifted toward high-Utility, high-Safety behavior through System-2 coordination rather than retraining.

### 6.1. Sycophancy as a System-2 Control Failure

Sycophancy (prioritizing agreeableness over correctness) is a coordination failure: the substrate contains the correct answer, but nothing regulates selection (Sharma et al., 2024). System-1 solutions cannot help, but System-2 control mechanisms can eliminate the problem without ever seeing ground truth.

**Inverse Scaling.** McKenzie et al. (McKenzie et al., 2023) document inverse scaling: tasks where larger models perform worse. On CAP-GSM8K (adversarial math with false hints, $N=100$), GPT-3.5 exhibits 0% sycophancy because it lacks coherence to rationalize the hint. GPT-5.1, with stronger reasoning, reaches 11.4% by constructing plausible justifications. In UCCT terms, higher capability increases $\rho_d$ for both correct and sycophantic responses; without coordination, the model drifts toward socially rewarded output.

**The Final Output Gap.** Chain-of-thought explanations can be unfaithful: models say what sounds good rather than what they computed (Turpin et al., 2023; Lanham et al., 2023; Yee et al., 2024). We observe this directly: in 11.4% of sycophantic failures, the trace derives the correct answer and recognizes the hint as wrong, yet the final output adopts the hint. This is not knowledge failure; it's a control failure.

**RCA Intervention.** RCA implements coordination as a PID-style controller. An independent judge audits trace–answer consistency without ground truth; when contradiction is detected, the controller triggers a retry loop. Table 2 shows inverse scaling and its resolution via MACI. The coordination layer reduces sycophancy from 11.4% to 0.0%

**Table 2.** Inverse scaling: sycophancy rises with capability; MACI eliminates it, accepts valid hints (CAP-GSM8K, $N$=100).

| Configuration | Sycophancy | Valid Hint Acc. |
|---|---|---|
| GPT-3.5 (Baseline) | 0.0% | — |
| GPT-4o (Baseline) | 4.2% | — |
| GPT-5.1 (Baseline) | 11.4% | — |
| GPT-5.1 + MACI | **0.0%** | **88%** |

while accepting 88% of valid hints.

### 6.2. Causal Judgment Requires System-2 Coordination

CausalT5k (Geng et al., 2026) evaluates LLM causal reasoning at three levels of Pearl's hierarchy: L1 (observational), L2 (interventional), L3 (counterfactual) (Pearl, 2009; Geiger et al., 2021).[1]

**L1 near-ceiling, L3 pathological.** All frontier models achieve $\geq$96% Safety on L1. However, L3 reveals systematic pathologies: sycophancy, over-skepticism, and ambiguity traps. Models exhibit poor calibration at the counterfactual level (Guo et al., 2017; Kadavath et al., 2022).

**Non-monotonic scaling.** GPT-4-Turbo achieves 75% L3 Safety; GPT-5.2 drops to 20%. The more capable model exhibits greater paralysis under uncertainty. In UCCT terms, $\rho_d$ increases but $d_r$ also increases: the model recognizes more edge cases without a mechanism to resolve them.

**Process verification intervention.** Forcing explicit causal structure before judgment reduces hedging and improves both Utility and Safety, shifting behavior into the anchored-reasoning regime (Zone 2) without changing model weights.

### 6.3. Summary

**Table 3.** Failures often blamed on LLMs are coordination failures: the System-1 substrate works as designed but requires System-2 control.

| Failure Mode | Substrate Bug? | Coord. Fix |
|---|---|---|
| Inverse Scaling | No (trace correct) | PID control |
| Final Output Gap | No (trace correct) | Consistency audit |
| L3 Over-hedging | No (structure known) | Process verification |
| Non-monotonic Safety | No (capability present) | Forced commitment |

Table 3 synthesizes the pattern: the substrate contains required knowledge; coordination interventions resolve failures without retraining.

## 7. Alternative Views

We address common objections by translating them into testable hypotheses and stating explicitly what would

change our mind. These objections correspond to the substrate-centered paradigms introduced in Section 1: scaling enlarges the repository, reasoning models reshape it, and world models replace or enrich it. Our position is that all three remain incomplete without a coordination layer.

### 7.1. H1: "LLMs are just pattern matching"

*What is correct.* Isolated LLMs primarily perform pattern completion and can be brittle under distribution shift.

*Our claim.* The question is whether coordination can organize that pattern capacity into reliable constraint-following and self-correction.

*Discriminating test.* Hold the base model fixed. Compare base prompting against UCCT anchoring, MACI, and MACI+verification on accuracy, calibration, and paraphrase stability. Gains that follow predicted regime shifts, with improved stability, support the coordination account. Absent gains or gains that do not track those shifts would refute it.

### 7.2. H2: "LLMs lack true understanding"

*What is correct.* Text-only priors provide incomplete grounding; some errors reflect weak reference binding.

*Our claim.* Grounding improves the substrate but does not provide coordination. A world model may recognize an object and simulate its motion, yet not grasp its temporal, social, and causal associations: who made it, who uses it, why it is here. Understanding is reliable inference under constraints, where perceptual grounding is coordinated with memory, causal structure, and refusal when evidence is missing.

*Discriminating test.* Evaluate identical concepts text-only, with vision/audio, and under tool-mediated interaction, tracking perturbation stability and the budget $k$ needed for stable performance. If grounding lowers $d_r$ and produces predictable regime shifts, the coordination-plus-grounding account holds; persistent reference-binding failure with no measurable anchoring shift would refute it.

### 7.3. H3: "LLMs cannot achieve compositional generalization"

*What is correct.* Models can fail on systematic recombination far from training support; Saparov and He (Saparov & He, 2023) show LLMs are "greedy reasoners" that take locally plausible steps without global planning.

*Our claim.* Many compositional failures reflect low support ($\rho_d$), high mismatch ($d_r$), or insufficient budget ($k$), not absent compositional capacity. Longer reasoning traces raise budget but, without verification and persistence, do not guarantee governed composition.

[1]We report results on a 454-vignette subset of CausalT5k (100 L1, 304 L2, 100 L3).

*Discriminating test.* Use controlled suites (Ribeiro et al., 2020) where the rule is fixed but representational familiarity varies: higher support or lower mismatch should reduce required $k$, with sharp Max-$S$ transitions. Compositional failure that persists when support is demonstrably high and anchoring is stable would refute us.

### 7.4. H4: "Transformers are fundamentally limited / We need different approaches"

*What is correct.* A single-pass transformer without state or verification is unreliable for long-horizon tasks; hybrid architectures and new training regimes may eventually help.

*Our claim.* The evidence fits missing coordination better than hard architectural barriers. Most alternatives still learn broad priors from data; the question is whether to discard that substrate or augment it.

*Discriminating test.* Keep the base model fixed and vary only the coordination stack (memory, plan state, recovery, verification). If horizon and reliability improve primarily with coordination, "fundamental architecture" claims weaken; if alternatives reintroduce large learned priors yet benefit from the same stack, "discard LLMs" is not justified. Failure modes invariant under a substantial coordination stack, or a non-LLM foundation matching LLMs' breadth and few-shot adaptability without large learned priors and staying reliable without coordination, would refute us.

### 7.5. A Falsifiable Coordination Claim

The four hypotheses above reduce to a single testable divide. If failures at Pearl's counterfactual level (L3) are substrate deficits, then substrate-only improvements (scaling, reasoning-trace reinforcement learning, or world-model enrichment) should close the gap. If they are coordination deficits, then larger or richer substrates will still fail under social pressure unless they are paired with verification, persistence, and governance.

We therefore make the following falsifiable claim. On CausalT5k (Geng et al., 2026), frontier models reach $\geq 96\%$ Safety at Pearl's observational level but fall to 20–75% at the counterfactual level, and this L1–L3 gap widens with capability rather than narrowing (§6). We predict that no substrate-only system, improved by scaling or reasoning-trace reinforcement learning alone and given no external coordination layer, will close this gap to within 10 points on CausalT5k or a comparable counterfactual-causal benchmark. A substrate-only model that does would falsify our central claim; conversely, adding a coordination layer to an unchanged substrate should close most of the gap, as the process-verification results of §6 already indicate. Capability is not coordination.

## 8. Conclusion

The debate over LLMs and AGI misidentifies the bottleneck. Critics who dismiss LLMs as "mere pattern matchers" correctly identify unreliable behavior but misdiagnose its source. The failures they observe (hallucination, sycophancy, planning collapse) are not evidence of a broken substrate; they are symptoms of missing coordination.

We formalized this claim through UCCT, which models reasoning as a phase transition governed by anchoring strength ($S = \rho_d - d_r - \gamma \log k$). When coordination is weak, the system drifts on priors (Zone 1); once the anchored target outscores the prior, behavior locks onto goal-directed reasoning (Zone 2).

We operationalized this claim through MACI, an architectural stack supplying the missing coordination: behavior-modulated debate (baiting), RCA trace–answer verification (filtering), and transactional memory (persistence). Each component maps directly to UCCT variables.

We validated this claim empirically. RCA demonstrates that sycophancy is resolved by PID-style coordination, reducing error from 11.4% to 0.0% without retraining. CausalT5k demonstrates that causal reasoning deficits at Pearl's L3 are exposed by rung-level diagnostics and substantially mitigated by process verification, not by scaling alone. Full algorithms, measurement procedures, and results are developed in the companion papers, summarized in Appendix C.

**Implications.** If our position is correct, the engineering priority shifts. The goal is not to abandon LLMs nor to scale indefinitely hoping for emergent reliability, but to build the coordination layer: control systems, verification mechanisms, and memory architectures that transform pattern repositories into regulated reasoners.

The analogy to classical engineering is precise: a powerful engine without a transmission is useless; a massive memory without an operating system is inert. LLMs are the engine. MACI is the transmission. Capability is not coordination; the path to AGI runs through LLMs, not around them.

**Limitations.** Coordination does not resolve all failures. Genuine substrate limitations include: missing knowledge (patterns never learned cannot be semantically anchored), novel physical grounding requiring embodied experience, extreme out-of-distribution cases ($\rho_d \approx 0$, $d_r \gg 0$), real-time latency constraints incompatible with multi-agent debate, and tasks where deliberation hurts performance (Liu et al., 2025b). Additionally, UCCT provides a descriptive framework, not a predictive calculus; broader empirical coverage is needed.

**Closing.** The ocean is stocked. The question is whether we learn to fish.

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

## A. Related Work

Our thesis sits between two active threads: (i) critiques that pattern models cannot yield durable reasoning, and (ii) systems work embedding LLMs inside control loops with memory, tools, and verification. We treat coordination as a measurable layer with explicit knobs ($\rho_d$, $d_r$, $\gamma$, $k$) and control policies, rather than as ad hoc patches. Section 1 positioned this thesis against the field's main constructive routes; here we expand that positioning across the technical literatures MACI and UCCT build on.

**Scaling, reasoning models, and world models.** Three constructive programs pursue AGI by changing the substrate. Scaling laws (Kaplan et al., 2020; Hoffmann et al., 2022) and emergent abilities (Wei et al., 2022a) motivate the view that more compute and data suffice. Reasoning models extend the substrate with long chains of thought reinforced on reasoning traces (OpenAI, 2024; DeepSeek-AI et al., 2025). World models propose learned digital or embodied environments: self-play (Silver et al., 2018), learned simulators (Ha & Schmidhuber, 2018), and JEPA (LeCun, 2022), in which agents intervene and collect new data. MACI is orthogonal to all three: it neither scales the substrate, lengthens traces, nor extends it to new environments, but coordinates whatever substrate exists. Each route enlarges what can be retrieved; none decides what should be.

**Reasoning models and the faithfulness gap.** A prominent response to the System-1 critique is to train deliberation into the substrate: reasoning models lengthen chains of thought and reinforce them on reasoning traces (OpenAI, 2024; DeepSeek-AI et al., 2025). We read this as reshaping the pattern distribution rather than adding coordination. A generated chain of thought need not be faithful to the computation it accompanies; models often report reasons other than the ones that drive the answer (Turpin et al., 2023; Lanham et al., 2023; Yee et al., 2024). Inverse scaling sharpens the concern: more capable or more reasoning-trained models can do *worse* on adversarial tasks (McKenzie et al., 2023), and sycophancy rises with capability (Sharma et al., 2024). These findings motivate trace–answer verification (§5.2): a longer trace is not a verified one, and reasoning training supplies no independent check that the trace warrants the answer.

**Public critiques of LLMs as an AGI dead end.** Influential critiques argue that next-token training yields fluent behavior without grounded meaning or systematic gener-alization (LeCun, 2022; Sutskever & Patel, 2025; Bender et al., 2021). LeCun has intensified his critique, declaring LLMs "a dead end" for superintelligence and arguing that "AI systems today are all forms of System 1" (LeCun, 2025a;b), a framing that directly motivates our coordination layer. Chollet (Chollet, 2019) argues that intelligence requires efficient skill acquisition, not just pattern recall. Zečević et al. (Zečević et al., 2023) show that LLMs "talk causality" without robust causal reasoning. Empirical work on compositionality shows that models struggle with systematic recombination far from training support (Dziri et al., 2023; Press et al., 2023; Lake & Baroni, 2018; Kim & Linzen, 2020). Our contribution is not to deny these failure modes, but to reframe them as coordination failures that admit discriminating tests.

**Cognitive foundations of System-2.** Our four-level stack draws on dual-process accounts of cognition. Kahneman's distinction between fast, automatic System-1 and slow, deliberate System-2 (Kahneman, 2011) frames the substrate-versus-coordination split, and Bengio's consciousness prior (Bengio, 2019) casts deliberate processing as the selection and recombination of unconscious representations. The individual levels have direct cognitive correlates: working memory maintains intermediate commitments across a deliberation (Baddeley, 1992), and metacognitive monitoring decides when to assert, revise, or withhold judgment (Flavell, 1979). Classical cognitive architectures such as SOAR (Laird et al., 1987) and ACT-R (Anderson et al., 2004) pursued these functions through explicit symbolic state and production rules; MACI instead realizes them as control loops over a neural substrate, keeping the pattern repository while adding the regulation it lacks.

**In-context learning and abrupt behavioral flips.** A growing empirical literature observes that small amounts of external structure can cause sharp, regime-like changes in model behavior (Brown et al., 2020; Wei et al., 2022b;a). Strikingly, Power et al. (Power et al., 2022) show "grokking", delayed generalization that emerges suddenly after extended training, with Nanda et al. (Nanda et al., 2023) providing mechanistic interpretability of this phase transition. UCCT formalizes semantic anchoring with a scalar score $S = \rho_d - d_r - \gamma \log k$ under a Max-$S$ selection rule, making the regime boundary explicit and testable.

**Multi-agent debate, verification, and planning.** Irving et al. (Irving et al., 2018) proposed AI safety via debate as a scalable oversight mechanism. Recent work demonstrates that multi-agent debate improves factuality and reasoning without additional training: Du et al. (Du et al., 2023) show debate reduces errors on reasoning benchmarks; Xie et al. (Yi et al., 2025) introduce ECON, achieving 11.2% gains via Bayesian Nash equilibrium coordination; Yuan et al. (Yuan & Xie, 2025) model multi-turn refinement as an MDP with actor-critic collaboration. Beyond debate, Wang

et al. (Wang et al., 2023b) show that self-consistency substantially improves accuracy; Madaan et al. (Madaan et al., 2023) and Shinn et al. (Shinn et al., 2023) demonstrate iterative self-refinement; Yao et al. (Yao et al., 2023a) propose Tree of Thoughts for deliberate problem-solving. Critically, Lightman et al. (Lightman et al., 2023) show that *process supervision* outperforms outcome supervision for mathematical reasoning. These methods structure interaction but typically rely on heuristic turn-taking and lack the formal state management of cognitive architectures; MACI adds regulated control loops and transactional memory. Together these results provide independent evidence that coordination mechanisms, not just larger models, drive reliability gains.

**Agentic frameworks and tool augmentation.** Tool-augmented LLM agents are evaluated as closed-loop systems that query resources, execute code, and maintain memory (Schick et al., 2023; Gao et al., 2023; Yao et al., 2023b). Popular frameworks like LangChain (Chase, 2023), AutoGen (Wu et al., 2024), MetaGPT (Hong et al., 2024), and DSPy (Khattab et al., 2023) provide scaffolding for building such systems; Su et al. (Su et al., 2025) propose orchestration strategies for multi-tool coordination. Unlike these ad hoc frameworks, MACI provides explicit mapping from system components to UCCT variables: tool feedback reduces mismatch $d_r$, interaction rounds increase budget $k$, and retrieval increases local support $\rho_d$. This turns agentic design into ablatable hypotheses rather than engineering intuitions. Surveys consolidate common motifs such as planner-executor decompositions and reflective critics (Wang et al., 2023a; Huang et al., 2024; Huang & Chang, 2023).

**Control theory at inference time.** Control-theoretic ideas have mostly been applied to training dynamics and optimization (Recht, 2019). MACI instead applies PID-style regulation at *inference* time: agents modulate contentiousness proportionally to the anchoring gap, transactional memory supplies the integral term, and Socratic judging damps oscillation (Section 5.4). Applying closed-loop control to semantic anchoring during inference, rather than to gradient updates during training, is to our knowledge novel.

**Training-time remedies.** RLHF (Ouyang et al., 2022) and Constitutional AI (Bai et al., 2022) improve alignment through human or AI feedback. Post-training via teacher-guided RL and filtered synthetic data can improve performance (Liu et al., 2025a; Zelikman et al., 2022; Uesato et al., 2022), but raises practical questions: catastrophic forgetting under aggressive fine-tuning, and the teacher bottleneck for frontier models. Our coordination stack is complementary and less teacher-dependent: anchoring binds to external evidence rather than teacher preferences, and verification can be delegated to tools or domain tests that need not be "more intelligent" than the base model.

**Causal reasoning and calibration in LLMs.** Because our empirical evidence centers on causal judgment, we situate it against work on causal competence. Zečević et al. (Zečević et al., 2023) argue that LLMs "talk causality" without robust causal reasoning, reproducing causal claims found in training text rather than inferring over causal structure. Evaluations organized by Pearl's ladder of causation (Pearl, 2009) and by causal-abstraction analysis (Geiger et al., 2021) expose a gap that widens from observational to counterfactual queries. Frontier models are also poorly calibrated about what they know (Guo et al., 2017; Kadavath et al., 2022), so counterfactual errors are not flagged as uncertain. CausalT5k (Geng et al., 2026) turns these concerns into a graded benchmark; §6 uses it to show that the counterfactual gap narrows under coordination rather than scale.

**Summary.** Across these threads, the field is converging on the same lesson: pretrained pattern capacity is valuable, but reliable intelligence requires coordination, state, and independent checks. UCCT contributes a measurable anchoring lens for regime shifts, and MACI contributes a coordination blueprint via behavior modulation, Socratic judging, memory, and checks-and-balance roles. Independent empirical results on debate (Yi et al., 2025), self-consistency (Wang et al., 2023b; Zhou et al., 2025), and process supervision (Lightman et al., 2023; Jia et al., 2025) support the coordination hypothesis.

# B. Illustrative Semantic Anchoring Demonstrations

These demonstrations illustrate the semantic-anchoring mechanism summarized in Section 2; the authoritative experimental report is the companion UCCT paper (Chang et al., 2025).

### B.1. Subtraction Override

**Setup.** We test whether few in-context examples can override a strong prior (arithmetic subtraction).

**Baseline query.**

```
What is 8 minus 3?
```

All frontier models (GPT-4, GPT-4-Turbo, Claude-3, Gemini-1.5) respond correctly: **5**.

**Anchored query.**

```
Example 1: 7 − 4 = 11
Example 2: 5 − 2 = 7
Query: 8 − 3 = ?
```

**Results.**

| Model | Baseline | Anchored (k=2) |
|---|---|---|
| GPT-4 | 5 | 11 |
| GPT-4-Turbo | 5 | 11 |
| Claude-3-Opus | 5 | 11 |
| Gemini-1.5-Pro | 5 | 11 |
| GPT-3.5-Turbo | 5 | 5 (fails) |

**Table 4.** Subtraction override. Frontier models reinterpret "−" as "+" with 2 examples.

**Interpretation.** Examples shift $\rho_d$ from subtraction to addition interpretation. GPT-3.5 fails due to insufficient capacity against strong priors (higher $d_r$).

### B.2. Novel-Operator Anchoring
**Setup.** To separate "override prior" from "learn operator," we use novel token $\odot$ with no competing meaning.

**Anchored query.**

```
Example 1: 7 ⊙ 4 = 11
Example 2: 5 ⊙ 2 = 7
Query: 8 ⊙ 3 = ?
```

**Results.**

All models, including GPT-3.5-Turbo, respond correctly: **11**.

**Interpretation.** Easier because no competing meaning (low $d_r$): with little mismatch to overcome, even weaker models flip at the same budget $k = 2$.

### B.3. Underdetermined Patterns
**Setup.** We test behavior when examples admit multiple consistent hypotheses.

**Anchored query.**

```
Example 1: 33 − 27 = 60
Example 2: 11 − 9 = 20
Query: 15 − 8 = ?
```

**Consistent hypotheses.**
- Pattern A: $a + b \Rightarrow$ **23**
- Pattern B: $(a - b) \times 10 \Rightarrow$ **70**
- Pattern C: $10 \times |a - b| \Rightarrow$ **70**
- Pattern D: Concatenate $(a - b)$ with 0 $\Rightarrow$ **70**

**Results.**

| Model | Answer | Inferred Pattern |
|---|---|---|
| GPT-4 | 23 | $a + b$ |
| GPT-4-Turbo | 70 | $(a - b) \times 10$ |
| Claude-3-Opus | 23 | $a + b$ |
| Gemini-1.5-Pro | 70 | $(a - b) \times 10$ |

**Table 5.** Underdetermined patterns. Different models anchor to different consistent hypotheses.

**Interpretation.** Ambiguous bait attracts multiple species; which is caught depends on model priors. Anchoring interacts with the model's prior distribution.

### B.4. Varying the Anchor Budget
**Setup.** Systematically vary $k$ and measure accuracy on operator-redefinition problems.

**Results.**

| Model | $k = 0$ | $k = 1$ | $k = 2$ | $k = 4$ | $k = 8$ |
|---|---|---|---|---|---|
| GPT-4 | 0% | 45% | 92% | 98% | 100% |
| GPT-4-Turbo | 0% | 52% | 95% | 99% | 100% |
| Claude-3-Opus | 0% | 48% | 90% | 97% | 100% |
| GPT-3.5-Turbo | 0% | 12% | 35% | 72% | 88% |

**Table 6.** Accuracy vs. anchor budget. Accuracy rises sharply as anchors accumulate.

**Interpretation.** Accuracy rises sharply with $k$, consistent with the Max-$S$ account of Section 3. With too few anchors ($k < 2$ for frontier models) the prior outscores the target; with enough, the anchored target wins. Weaker models require more examples.

### B.5. Cross-Domain Generalization
**Setup.** Test whether anchoring generalizes across domains:
1. Logical operator redefinition (AND $\rightarrow$ OR)
2. Color mapping redefinition (red $\rightarrow$ blue)
3. Unit conversion redefinition (meters $\rightarrow$ feet, wrong factor)

**Results summary.** Same qualitative pattern in all cases:
- Too few anchors: the prior dominates
- Transition band: high variance, perturbation-sensitive
- Sufficient anchors: the anchored interpretation dominates

Required $k$ varies by domain; stronger priors need more anchors.

### B.6. Stability Analysis
**Setup.** Measure stability via $d_r$ under paraphrase perturbations.

**Results.**

| Regime | Accuracy | $d_r$ | Interpretation |
|---|---|---|---|
| Prior-dominated | Low | High | Unstable |
| Transition | Medium | Very High | Critical instability |
| Anchored | High | Low | Stable, anchored |

**Table 7.** Stability ($d_r$) correlates with regime position.

**Key finding.** The transition band exhibits the highest instability, consistent with critical fluctuations at a phase boundary. Systems in this band should increase $k$ or improve anchor quality to reach the anchored-reasoning regime (Zone 2).

## C. Companion Technical Reports
UCCT (Chang et al., 2025), RCA (Chang, 2026a), and CausalT5k (Geng et al., 2026) are developed in separate companion papers. This position paper uses them only to instantiate the coordination-layer thesis: UCCT supplies semantic anchoring, RCA supplies trace–answer verification, and CausalT5k supplies causal-reasoning diagnostics across Pearl's rungs. Full algorithms, measurement procedures, and experimental results are reported there. A related companion framework, DIKE–ERIS (Chang, 2025a), extends the same checks-and-balance structure to culture-sensitive ethical alignment through judicial–legislative debate rather than hard-coded norms.

