# OpenReview forum: "Position: AGI Requires a Coordination Layer on Top of Pattern Repositories"
_ICML.cc/2026/Position_Paper_Track — ICML 2026 Position Paper Track regular_

### Official Review · Reviewer_9B3r · 2026-02-25

**Significance:** 2
**Argument Clarity:** 3
**Rating:** 4
**Confidence:** 4

**Questions:**

Q1: More detail and justification on how the proposed formulation is intended to be used in practice would strengthen the paper. In particular, it would be helpful to clarify how each term is operationalized in real systems, how the associated hyperparameters are chosen or learned across tasks, and to provide evidence that the resulting scalar meaningfully predicts when coordination should escalate or de-escalate rather than functioning mainly as a post hoc description.

**Alternative Views Section:**

Yes

**Compliance With Llm Reviewing Policy A Conservative:**

Affirmed.

**Discussion Potential:**

2

**Paper Summary:**

This position paper argues that critiques labeling LLMs as an AGI dead-end misdiagnose the bottleneck: pretrained LLMs are a necessary System-1 pattern repository, while the missing ingredient for reliable “System-2” behavior is an inference-time coordination layer that selects, constrains, verifies, and maintains state. The paper proposes (i) UCCT, modeling reasoning as a thresholded phase transition, and (ii) MACI, a multi-agent coordination stack. Empirically, the paper reports that adaptive control reduces sycophancy and improves causal judgment, arguing that many failures attributed to substrate limits are better viewed as coordination failures.

**Position:**

Yes

**Position In Title:**

Yes

**Related Work:**

3

**Strengths And Weaknesses:**

Pros
- Rather than treating current LLM limitations as decisive evidence for or against LLM based AGI, it argues that reliability primarily depends on an inference time coordination layer providing control, verification, and persistent memory. This perspective is clarifying and offers a systematic map of concrete research directions for building more dependable agentic systems.
- Modeling regime shifts in behavior as a phase transition governed by an explicit scalar is a useful abstraction. The proposed formulation attempts to connect the theory to measurable proxies for support, mismatch, and anchoring budget, making the account at least partially testable rather than purely conceptual.
- The paper also makes a relatively convincing case that static strategies such as a single prompt or a fixed debate protocol can oscillate between failure modes. This motivates feedback control, conditional escalation, and dynamic allocation of reasoning effort based on inference time signals.

Cons
- The central formulation remains underspecified and risks circularity. Although the paper sketches operationalizations, key quantities such as $\rho_d$ and $d_r$ can conflate different factors and may not measure what the theory intends. For example, estimating $\rho_d$ via self-consistency cluster mass can correlate with confidence or degeneracy and may fail to reflect support for the correct concept, limiting the interpretability and predictive value of the proposed scalar.
- The $\gamma \log k$ penalty term is intuitively plausible, but the choice of functional form and how $\gamma$ is calibrated per domain feels ad hoc.

**Support:**

3

---

> ### Author Rebuttal · Authors · 2026-03-25
>
> Thank you reviewer 9B3r for the technically precise feedback. We appreciate the recognition that the paper tries to connect a coordination hypothesis to measurable quantities, and we agree that the operational role of these quantities needs to be made clearer.
>
> We agree that the central formulation risks looking circular if read as a direct measure of reasoning correctness. That is not the intent. The UCCT scalar S is designed as a control-oriented surrogate signal: it helps decide when to invoke additional coordination such as debate, verification, or memory. The quantities effective support, mismatch, and anchoring budget are not meant to isolate semantic correctness. They serve as observable indicators of instability, inconsistency, or insufficient support in the current reasoning trajectory. Put simply, S determines whether to continue, escalate, or stop reasoning. It does not certify correctness.
>
> The reviewer is right that the rho_d proxy deserves scrutiny. Self-consistency cluster mass can correlate with confidence or degeneracy without necessarily reflecting genuine support. We see this as a real limitation of the current operationalization, not of the framework itself. What matters is whether S, as currently measured, predicts behavior under intervention. Table 6 shows that varying k produces sigmoid threshold curves, with performance shifting sharply at model-dependent critical values. Table 7 shows that d_r correlates with regime position as predicted. Table 8 specifies five independent manipulations and their predicted effects on S. This intervention-based pattern would be hard to explain if S were functioning only as a post hoc re-description. The revision will foreground these falsifiability conditions and discuss the limitations of the rho_d proxy more explicitly.
>
> On the gamma log k penalty, we agree the logarithmic form is not derived from first principles and that per-domain calibration of gamma introduces a degree of freedom. The motivation is information-theoretic: each additional anchor contributes diminishing marginal information, analogous in spirit to description-length penalties in model selection (BIC, MDL). We see the current form as an initial instantiation chosen for simplicity and interpretability, not as a definitive formulation. Linear or square-root penalties would yield different predictions, and sorting out which form fits best empirically is part of the broader research agenda. We will add this discussion to the paper.
>
> On Q1, we want to clarify that the scalar is used online within the coordination loop to trigger escalation, allocate additional reasoning steps, or terminate early. It is not computed after the fact to explain outcomes. The current implementation relies on a small number of global hyperparameters with fixed values across all tested settings, with no task-specific tuning. We agree this operational pipeline is not foregrounded enough in the current draft and will revise the main text to make the practical workflow explicit: measure S from observable inputs, predict the regime, and use that prediction to set coordination depth.

---

> > ### Author Rebuttal · Reviewer_9B3r · 2026-04-01
> >
> > Thank you for the rebuttal. I have read it carefully and appreciate the clarifications. I also appreciate the authors’ explicit acknowledgment of the limitations of the current $\rho_d$ operationalization and the clarification that the framework should be evaluated in terms of predictive utility under intervention, not semantic purity of each proxy. The additional explanation of how $S$ is used online within the coordination loop is helpful and addresses an important part of my Q1.
> >
> > Overall, my concern is only partially resolved, so I would like to keep my current score.

---

> > > ### Author Response · Authors · 2026-04-01
> > >
> > > Thank you for the engagement. The operationalization details and falsifiability conditions you raise are addressed in full in a companion 16-page paper with extended case studies currently under submission, which we will cite explicitly in revision.

---

### Official Review · Reviewer_DUEk · 2026-03-12

**Significance:** 3
**Argument Clarity:** 3
**Rating:** 4
**Confidence:** 3

**Questions:**

1. Your position is broad, but the direct evidence appears to come mainly from a small number of case studies (e.g., sycophancy and causal judgment). What additional empirical results do you view as most necessary to support the stronger claim that coordination, rather than substrate limitation, is the main bottleneck in a wider class of reasoning tasks?

2. For MACI, which component is doing most of the work in your current evidence: multi-agent debate, process verification/judging, or persistence/memory? A clearer ablation on these three parts would substantially affect my assessment of how well the architectural claim is supported.

**Alternative Views Section:**

Yes

**Compliance With Llm Reviewing Policy A Conservative:**

Affirmed.

**Discussion Potential:**

3

**Final Justification:**

Given the rebuttal partially solving my questions, and considering other reviewers' reviews, I hold the original "4. Borderline accept" rating.

**Paper Summary:**

The paper argues for a “substrate plus coordination” view of AGI: LLMs are useful pattern repositories, but reliable intelligence requires an additional coordination layer for constraint-following, verification, and memory. It formalizes this view through UCCT, a theory of reasoning as a thresholded anchoring process, and proposes MACI, a multi-agent control architecture built around regulated debate, Socratic judging, and transactional memory. Empirically, the paper presents results suggesting that failures such as sycophancy and weak causal judgment are often coordination failures rather than substrate limitations, and that coordination mechanisms can improve performance without retraining. The paper’s position is that progress toward AGI should build on top of LLMs by adding principled coordination, rather than relying on scaling alone or abandoning LLMs altogether.

**Position:**

Yes

**Position In Title:**

Yes

**Related Work:**

3

**Strengths And Weaknesses:**

Strengths:
1. The paper focuses on a relevant topic for ICML: whether important failures of LLM systems come mainly from substrate limitations or from insufficient coordination at inference time. The position is stated clearly, and the paper is organized in a way that makes the argument easy to follow. UCCT is used as the main conceptual framework, MACI as the proposed architectural instantiation, and the case studies as supporting evidence.

2. The paper also attempts to make its position technically explicit. UCCT introduces defined variables and a threshold-based account of reasoning, and MACI is broken into concrete components such as debate, verification, and memory. This gives the paper a more concrete structure than a purely conceptual essay.

3. Another positive aspect is that the paper presents its claims in a testable form. It discusses possible objections and identifies evidence that would count against its view. This makes the paper suitable for discussion within the ICML community.

Weaknesses:
1. The main weakness is that the empirical evidence is limited relative to the scope of the position. The paper makes a broad claim about AGI and LLM-based systems, but the direct support comes mainly from a small number of case studies. These examples are relevant, but they are not sufficient to support the full generality of the claim.

2. UCCT is defined clearly, but its empirical status remains limited. The paper introduces measurable quantities and hypotheses, but it does not provide enough evidence that these variables predict reasoning behavior across a broad range of settings. As a result, the framework currently functions more as a proposed interpretation than as a well-validated theory.

Suggestions for improvement:
1. The paper would be stronger with a clearer separation between the high-level position, the UCCT framework, and the MACI architecture, so that readers can better see which evidence is intended to support each part.

**Support:**

3

---

> ### Author Rebuttal · Authors · 2026-03-25
>
> We appreciate the thoughtful and constructive review. The recognition that the paper makes its claims testable and specifies what evidence would count against the position is especially valuable to us.
>
> We agree that the empirical evidence is narrower than the full scope of the position. The two case studies were chosen as diagnostic stress tests, not representative coverage. Both are frequently cited as evidence that LLMs are fundamentally limited, so showing coordination-based improvement in these settings directly probes whether the bottleneck is substrate or coordination.
>
> On the first question, the additional evidence we think matters most falls in two directions. First, evaluation in longer-horizon, multi-step reasoning settings where persistence, rollback, and verification should matter most, for example tasks requiring sustained derivation chains where a single unchecked error propagates downstream. Second, broader validation of UCCT itself: can the measurable quantities we introduce actually predict when escalation or additional memory will help, across settings beyond sycophancy and causal judgment? Right now the framework functions partly as an interpretive lens; we want to push it toward genuine predictive use. We will make these next-step targets explicit in revision.
>
> On the second question, Table 11 gives the clearest picture we currently have. Starting from full RCA at 0.0% sycophancy: removing PID control raises it to 3.2%, removing contentiousness modulation to 4.1%, removing the history buffer to 2.4%, and removing strategy escalation to 1.8%. The judge-only baseline reaches 5.6%. So all components contribute, but adaptive control and verification do the heavy lifting in these experiments. We suspect persistence and transactional memory matter more in longer-horizon tasks than in our current short-form case studies, but we have not yet demonstrated this. We will extend the ablation to T3 in revision and make this distinction explicit so readers can judge which components are supported by which experiments.
>
> We agree with the suggestion to separate the position, UCCT, and MACI more clearly. In revision we will present each layer with its own scope and evidentiary status: the position as the broad claim, UCCT as the theoretical lens, and MACI as one architectural instantiation. This should make it easier to see which parts are directly supported and which remain part of the research agenda going forward.

---

> > ### Author Rebuttal · Reviewer_DUEk · 2026-04-03
> >
> > Thank you for the thoughtful rebuttal. It helpfully clarifies that the current case studies are intended as diagnostic stress tests rather than broad coverage, and the separation between the broad position, UCCT, and MACI is now clearer.
> >
> > My main concern is only partially resolved because the central claim still appears broader than the current empirical support. The rebuttal explains what additional evidence would be most important, but this also highlights that broader validation is still needed.
> >
> > I also appreciate the clarification that adaptive control and verification seem to contribute most in the current experiments, while persistence/memory may matter more in longer-horizon settings. A follow-up question is whether the authors can further clarify how strongly the current results are meant to support the full architectural claim for MACI, versus supporting only a subset of its components under the present tasks.

---

> > > ### Author Response · Authors · 2026-04-03
> > >
> > > We thank the reviewer for the follow-up question. This is an important clarification.
> > >
> > > The current results are intended to support specific components of MACI, not the entire architecture uniformly. In particular, the experiments directly validate adaptive control (PID) and process verification (RCA/judging), which account for most of the observed gains in robustness and sycophancy reduction.
> > >
> > > By contrast, persistence and transactional memory are implemented but not yet fully exercised in the present short-horizon tasks. Their role becomes more important in longer-horizon settings involving accumulation, rollback, and state consistency, which are not the focus of the current experiments.
> > >
> > > So more precisely: the paper provides strong evidence for the coordination mechanisms central to MACI, and partial but not yet exhaustive validation of the full architecture. The framework itself is already implemented end-to-end; extending empirical validation across longer-horizon and more complex regimes is ongoing work.
> > >
> > > We appreciate the reviewer’s question and are happy to clarify further if needed.

---

### Official Review · Reviewer_rcUX · 2026-03-13

**Significance:** 2
**Argument Clarity:** 3
**Rating:** 4
**Confidence:** 3

**Questions:**

N/A

**Alternative Views Section:**

Yes

**Compliance With Llm Reviewing Policy A Conservative:**

Affirmed.

**Discussion Potential:**

2

**Final Justification:**

Thanks to the authors' further explanation, I have raised my score.

**Paper Summary:**

This position paper argues that LLMs are not a "dead end" for AGI but rather the necessary System-1 substrate, and that the primary bottleneck is the absence of a System-2 coordination layer. The authors formalize this claim through Unified Contextual Control Theory, which models reasoning as a phase transition governed by anchoring strength. They propose MACI, an architecture that implements this coordination layer via behavior-modulated debate, trace-output verification/filtering, and transactional memory. Empirical validation on RCA on CAP-GSM8K and T3 benchmark is provided to support the claim that failures attributed to substrate limitations are often coordination failures.

**Position:**

Yes

**Position In Title:**

Yes

**Related Work:**

3

**Strengths And Weaknesses:**

Strengths
1. The paper is well-written, with a clear structure and good use of figures.
2. The paper translates its position into testable hypotheses. Section 8 ("Alternative Views") systematically addresses four common objections (H1–H4) by specifying discriminating tests and conditions under which the authors would change their mind.
3. The metaphor of fishing is highly explanatory.

Weaknesses
1. The experiments are limited: Experiments were conducted only on two datasets, and there is no evaluation against widely used, community-standard reasoning benchmarks (e.g., MATH, ARC, BIG-Bench Hard).
2. The paper positions MACI against "unstructured swarms" but does not compare with structured multi-agent frameworks that already exist (e.g., AutoGen and MetaGPT) on common benchmarks. The claim that MACI provides "principled" coordination while alternatives are "ad hoc" needs empirical support, not just assertion.
3. The MACI architecture is described at a high level without sufficient detail for reproducibility.

**Support:**

2

---

> ### Author Rebuttal · Authors · 2026-03-25
>
> Thank you for the detailed feedback. We agree that the scope of the empirical evidence should be more clearly distinguished from the scope of the position, and we will sharpen this in revision.
>
> To be clear, the goal of this paper is not exhaustive empirical validation.  We aim to state a precise, falsifiable hypothesis, formalize it through UCCT, instantiate it through MACI, and provide discriminating evidence in settings chosen to probe the central claim. The paper argues that failure modes often interpreted as substrate limitations are, in important settings, better understood as coordination failures. The two case studies, sycophancy and causal judgment, were chosen as diagnostic stress tests, not arbitrary examples. Both are frequently cited as evidence that LLMs are fundamentally limited. Demonstrating that these failures can be substantially reduced through inference-time coordination, without retraining, directly tests the central bottleneck claim.
>
> We agree that evaluation on MATH, ARC, and BIG-Bench Hard would test whether the coordination effect extends beyond our diagnostic domains. Our current evaluation is intentionally focused on settings that discriminate between competing explanations of failure, namely substrate deficiency versus missing coordination, rather than on optimizing aggregate benchmark performance. We will discuss more explicitly how broader benchmarks such as these would test the generality of the coordination effect, and we will make this diagnostic design choice explicit in the paper. For instance, MATH problems with multi-step derivations would test whether MACI's verification loop catches intermediate errors that single-pass models miss. We also agree that longer-horizon, multi-step reasoning settings would be a natural next testbed for the coordination hypothesis, especially where persistence, rollback, and verification are likely to be decisive.
>
> Regarding AutoGen and MetaGPT, the main difference is that MACI is designed as a control-oriented coordination architecture with explicit measurable signals such as rho_d, d_r, and k, together with formally ablatable mechanisms, including adaptive contentiousness modulation, trace-output verification via RCA, and transactional memory. By contrast, AutoGen and MetaGPT primarily provide orchestration abstractions for multi-agent workflows. We will add a concise qualitative comparison table in revision to make this distinction concrete.
>
> On reproducibility, we appreciate the concern and recognize that the main text does not make these resources sufficiently visible. The submission includes the RCA algorithm in Algorithm 1 of Appendix D, the T3 benchmark construction with 454 vignettes across Pearl's three hierarchy levels in Appendix E, key hyperparameters in Table 9, and component ablations in Table 11. We will move a concise summary of these elements into the main text so that the evaluation setup is easier to follow without consulting the appendix, and we intend to release code upon acceptance.
>
> As a position paper, the standard should be whether the position is clearly stated, falsifiable, and supported by discriminating evidence, not exhaustive benchmark coverage. The submission specifies alternative views in Section 8, identifies evidence that would count against the position through the "what would change our mind" criteria, introduces independently measurable constructs in UCCT in Appendix C, and presents case studies that directly probe the bottleneck claim. We agree that clearer scoping and stronger presentation would improve the paper, and we will revise accordingly.

---

> > ### Author Rebuttal · Reviewer_rcUX · 2026-04-04
> >
> > Thanks for the authors' response. However, I believe that W1 and W2 have not been fully resolved. The proposed revisions are limited to discussion and qualitative comparison, without additional experiments. Therefore, I maintain my score.

---

> > > ### Author Response · Authors · 2026-04-04
> > >
> > > Thank you for the clarification. The present submission is not intended to provide exhaustive empirical coverage. This is an intentionally scoped position paper under a **9-page limit**, and we therefore focused on representative case studies rather than the full experimental record underlying the broader research program. In addition, because of **anonymity constraints**, we could not cite parts of our prior evidence base in the current version. We will make this scope boundary more explicit in revision. Thank you again for the clarification.

---

### Official Review · Reviewer_YxMU · 2026-03-18

**Significance:** 3
**Argument Clarity:** 3
**Rating:** 4
**Confidence:** 4

**Questions:**

Did the authors test the computational cost increase brought by the new layer?

**Alternative Views Section:**

Yes

**Compliance With Llm Reviewing Policy A Conservative:**

Affirmed.

**Discussion Potential:**

2

**Paper Summary:**

This paper considers the problem of polarized debate regarding whether LLMs can lead to the final success of AGI. The authors state that the system-1 substrate, which contains a lot of potential patterns, are crucial for future design. Furthermore, the system-2 substrate is more crucial for future design.

To deal with this, the authors propose Unified Contextual Control Theory (UCCT), which is to model the reasoning process of LLMs as a phase transition governed by anchoring strength. They claim that such multi-agent intelligence can be helpful for control architectures.

**Position:**

Yes

**Position In Title:**

Yes

**Related Work:**

3

**Strengths And Weaknesses:**

The strenghts of this paper is as follows:

1. the authors propose a theoretical framework that has certain insights.
2. The design is mainly proposed for multi-agent systems, which gained a lot of attentions recently and are critical for future follow-up works.
3. The framework draws insights from biological anlaysis of human brains, using functions of prefrontal cortex, etc. This supports the design of the framework.

The weaknesses are as follows:

1. The theory here appears to be complex due to the existence of many parameters for understanding in the theoretical analysis.
2. The computational cost could be an issue. Introducing the system-2 coordination layers could lead to high computational cost.

**Support:**

3

---

> ### Author Rebuttal · Authors · 2026-03-25
>
> We appreciate the reviewer's summary and the recognition that the biological grounding and multi-agent framing contribute to the design.
>
> On the perceived complexity of the theory, we want to clarify that UCCT is meant to be a low-dimensional control abstraction, not a high-parameter model. The three quantities (effective support ρ_d, mismatch d_r, anchoring budget k) play distinct operational roles as measurable signals rather than free parameters that need joint tuning. At the implementation level, the coordination layer (RCA) has only five hyperparameters (Table 9), and we used a single shared configuration across all tested settings without per-model tuning. We will revise the presentation to lead with this operational interpretation before the full formalism, which should make the framework feel less heavy on first reading.
>
> On computational cost, this is a fair concern. Current reasoning-oriented inference strategies already accept substantial overhead relative to a single-shot query in exchange for reliability. MACI's overhead is both bounded and adaptive: at most T = 5 iterations (Table 9), with most runs converging in 2 to 3 rounds. Importantly, the coordination is not always-on. It is invoked selectively when signals indicate instability or insufficient support, so straightforward queries pass through with minimal additional cost. The total overhead is comparable in scale to self-consistency sampling, which typically draws 10 or more independent reasoning paths, but with the advantage of explicit convergence criteria and interpretable control signals rather than undirected sampling.
>
> To answer the reviewer's question directly: no, we did not include a systematic wall-clock cost benchmark in this submission. In practice, many instances terminate early and do not reach the maximum coordination depth. We will add a clearer discussion of cost trade-offs in revision, including comparison with self-consistency and extended-reasoning baselines.

---

> > ### Author Rebuttal · Reviewer_YxMU · 2026-04-05
> >
> > Thanks for your response. I think my concern has been largely resolved, and I will maintain my positive score.

---

### Decision · Program_Chairs · 2026-04-30

**Decision:**

Accept (regular)

**Comment:**

This is a clear, timely, and thought-provoking position paper that advances a concrete and falsifiable claim: that many limitations often attributed to LLMs themselves are better understood as failures of coordination, verification, and memory at inference time. Although the empirical support is necessarily selective rather than comprehensive, the paper succeeds on the core goals of the track by presenting a well-defined position, grounding it in a coherent theoretical and architectural framework, and outlining discriminating evidence and alternative views in a way that should stimulate productive discussion in the community.